# Gestational breast cancer in New South Wales: A population-based linkage study of incidence, management, and outcomes

Nadom Safi[1,2], Christobel Saunders[3], Andrew Hayen[1], Antoinette Anazodo[4,5], Kei Lui[4], Zhuoyang Li[2], Marc Remond[2], Michael Nicholl[6], Alex Y. Wang[1], Elizabeth Sullivan[1,2]*

1 School of Public Health, University of Technology Sydney, Sydney, NSW, Australia, 2 Faculty of Health and Medicine, University of Newcastle, Callaghan, NSW, Australia, 3 Faculty of Health and Medical Sciences, The University of Western Australia, Crawley, WA, Australia, 4 School of Women's and Children's Health, University of New South Wales, Sydney, NSW, Australia, 5 Nelune Comprehensive Cancer Centre, Prince of Wales Hospital, Randwick, NSW, Australia, 6 Faculty of Medicine and Health, The University of Sydney, Sydney, NSW, Australia

* e.sullivan@newcastle.edu.au

**Data Availability Statement:** Primary data cannot be shared publicly because they are confidential health data held by the New South Wales Ministry of Health and subject to Australian privacy

## Abstract

### Background

The incidence of gestational breast cancer (GBC) is increasing in high-income countries. Our study aimed to examine the epidemiology, management and outcomes of women with GBC in New South Wales (NSW), Australia.

### Methods

A retrospective cohort study using linked data from three NSW datasets. The study group comprised women giving birth with a first-time diagnosis of GBC while the comparison group comprised women giving birth without any type of cancer. Outcome measures included incidence of GBC, maternal morbidities, obstetric management, neonatal mortality, and preterm birth.

### Results

Between 1994 and 2013, 122 women with GBC gave birth in NSW (crude incidence 6.8/100,000, 95%CI: 5.6–8.0). Women aged ≥35 years had higher odds of GBC (adjusted odds ratio (AOR) 6.09, 95%CI 4.02–9.2) than younger women. Women with GBC were more likely to give birth by labour induction or pre-labour CS compared to women with no cancer (AOR 4.8, 95%CI: 2.96–7.79). Among women who gave birth by labour induction or pre-labour CS, the preterm birth rate was higher for women with GBC than for women with no cancer (52% vs 7%; AOR 17.5, 95%CI: 11.3–27.3). However, among women with GBC, preterm birth rate did not differ significantly by timing of diagnosis or cancer stage. Babies born to women with GBC were more likely to be preterm (AOR 12.93, 95%CI 8.97–18.64), low birthweight (AOR 8.88, 95%CI 5.87–13.43) or admitted to higher care (AOR 3.99, 95%CI 2.76–5.76) than babies born to women with no cancer.

regulations. Ethics approval for this project only authorizes specific researchers named in the original ethics application access to the de-identified linked data derived from the primary health data sets held by New South Wales Health. Data inquiries can be directed to: NSW Population & Health Services Research Ethics Committee. CINSW-Ethics@health.nsw.gov.au Phone Phone: 02 8374 5689 (Ethics Officer) or 02 8374 3610 (Executive Officer, Manager Research Ethics).

**Funding:** This study received funding from the following sources: Cancer Council NSW grant (reference RG 18-02, awarded to ES); University of Technology Sydney (UTS), Faculty of Health (awarded to ES); UTS Doctoral Scholarship (awarded to NS); and Australian Government Research Training Program (awarded to NS).

**Competing interests:** The authors have declared that no competing interests exist.

## Conclusion

Women aged ≥35 years are at increased risk of GBC. There is a high rate of preterm birth among women with GBC, which is not associated with timing of diagnosis or cancer stage. Most births followed induction of labour or pre-labour CS, with no major short term neonatal morbidity.

## Introduction

In 2018, breast cancer was the most commonly diagnosed cancer in women, globally representing 24.2% of all cancers in women, and the most common cause of cancer-related mortality in women [1]. In Australia, breast cancer is the second most common cancer diagnosed during pregnancy after melanoma, with an incidence of 7.3 per 100,000 women giving birth [2]. The incidence of gestational breast cancer (GBC), defined as a first-time diagnosis of breast cancer during pregnancy, is increasing in high-income countries, in part due to the increasing age of mothers [3–5].

Women with GBC have higher rates of adverse obstetric outcomes, including thromboembolic events, sepsis, induction of labour and pre-labour cesarean section [2, 6, 7]. Preterm birth has been identified as the main adverse neonatal outcome for babies born to women with GBC [7]. It has been reported that preterm birth is a risk factor for developmental problems, irrespective of whether or not a baby is born to a women with GBC [8, 9]. Decisions around preterm delivery in the majority of cases of GBC are taken without any obvious clinical indication [10]. This is concerning as it has been suggested that preterm birth is the main risk factor for developmental problems in babies born to women with GBC irrespective of whether or not they were exposed to chemotherapy during pregnancy [7].

Our study aimed to examine the incidence, timing of diagnosis, obstetric management and perinatal outcomes of women with a first-time diagnosis of breast cancer during pregnancy (GBC) and their babies in New South Wales (NSW), Australia. We also examined whether decisions to deliver preterm babies iatrogenically by labour induction or pre-labour caesarean section (CS) were associated with the timing of breast cancer diagnosis during pregnancy and/or the stage of cancer at diagnosis.

## Methods

We conducted a population-based cohort study using linked NSW Health data. The study population included all women with pregnancies that ended in live birth or stillbirth in NSW between 1 January 1994 and 31 December 2013. Birth was defined as the delivery of an infant of at least 400 grams birthweight or at least 20 weeks gestation whether live or stillborn [11]. For this study, gestational breast cancer (GBC) was defined as a first-time diagnosis of primary breast cancer during pregnancy.

The study group comprised all eligible pregnancies with GBC. The comparison group comprised women who delivered with no history of cancer before or during pregnancy (Fig 1).

We used three linked datasets: Perinatal Data Collection (PDC), the NSW Cancer Registry (NSWCR) and the Admitted Patient Data Collection (APDC). The PDC is a state-wide surveillance system that captures data relating to patterns of pregnancy care, services and outcomes for all births in NSW (whether in public or private hospitals or home births) [12]. The NSWCR is a population-based cancer registry that captures demographic, incidence, cancer stage and death information of all people diagnosed with cancer (excluding non-melanoma

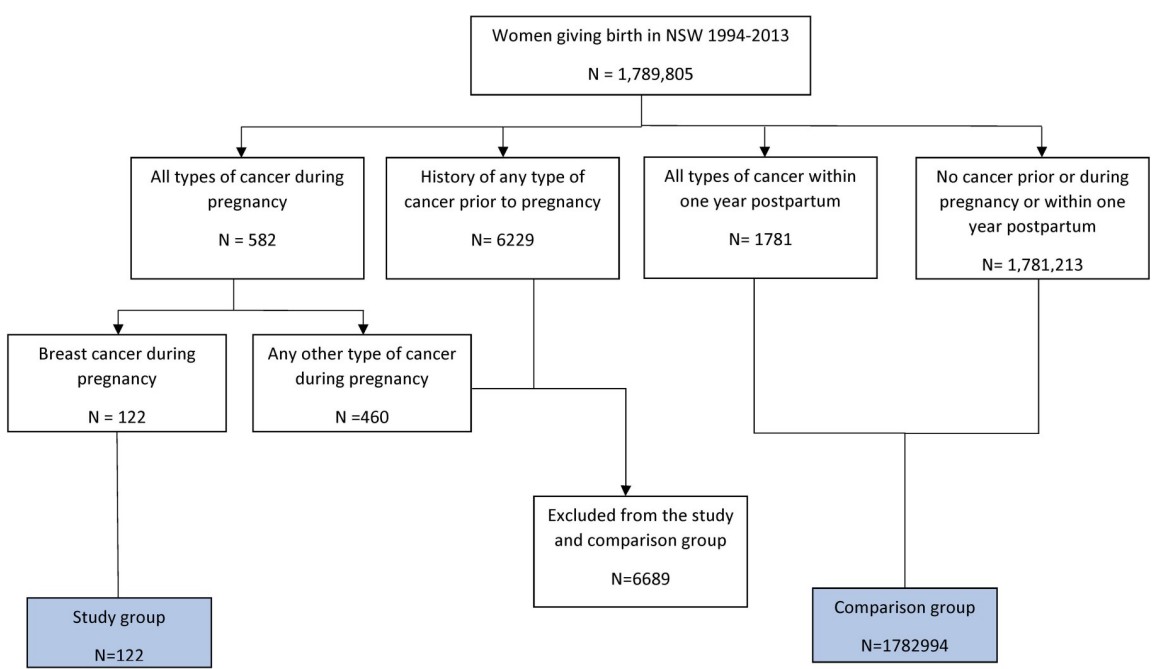

**Fig 1. Selecting the study and comparison groups.**

skin cancers) in NSW [12]. The APDC provides information on discharges, transfers or deaths on all patients admitted to all hospitals in NSW (public, private, repatriation and psychiatric hospitals). APDC data were only available from July 2001 [12]. The NSW Centre for Health Record Linkage (CHeReL) performed the data linkage. Details of the data linkage process are available on the CHeReL website [13]. The PDC was used as the primary dataset to identify the study cohort (NSW pregnancies from 1994 to 2013) while the NSWCR was used to identify the group of women with GBC in the study cohort. The APDC was merged based on the babies' Project-specific Person Numbers (PPN) [13] and was used to determine the frequency of hospital admissions and any diagnoses during the neonatal period for babies born to women with GBC.

Data relating to the degree of spread of cancer was obtained from NSWCR and was categorized as follows: stage 0—carcinoma in situ; stage 1—cancer localized to the tissue of origin; stage 2—cancer that has spread to the regional lymph nodes and/or adjacent organs; and stage 3—distant metastasis. We modified these data staging categories so as to reconcile them with the Royal College of Pathologists of Australasia cancer staging classification system. That is, final cancer staging was determined as follows: stage 1 cancer localized to the tissue of origin; stages 2–3 cancer that has spread to the regional lymph nodes and/or adjacent organs (the chest wall and/or the skin); stage 4 involves distant metastasis [12, 14]. Our dataset did not include information on stage 0 cancer, carcinoma in situ (CIS).

We classified women giving birth in NSW into three groups:

i. The study group (GBC group) comprised women with a first-time diagnosis of breast cancer during pregnancy;

ii. The comparison group comprised women without a history of cancer before or during pregnancy;

iii. An excluded group that comprised women with any type of cancer (including breast cancer) diagnosed prior to pregnancy (as prior cancer and its treatment may affect pregnancy outcomes [15, 16]) and women with cancer other than breast cancer diagnosed during pregnancy (as any decisions on their pregnancy management may not have differed from those for women with GBC).

## Main outcome measures

Maternal outcomes included pregnancy and birth management and complications (induction of labour, caesarean section (CS)), pregnancy complications (gestational diabetes and gestational hypertension) and maternal mortality. Neonatal outcomes included perinatal death (stillbirth or neonatal death), preterm birth (<37 weeks gestation), low birthweight (<2500 gm), small for gestational age (SGA) (birthweight below the 10th percentile for the age and sex [17]), intraventricular haemorrhage, and respiratory distress syndrome of newborn.

## Statistical analysis

The chi-squared test was used to compare the prevalence of SGA between preterm and term babies born to women with GBC. Mann–Whitney U test was used to examine the difference in median gestational age at birth between women with GBC and women with no cancer. Independent samples t-test was used to compare the mean difference in maternal age and baby birthweight between the study and comparison groups.

A Poisson regression model was used to examine the estimated increase in the incidence of GBC each year. The indirect age-standardized rate was used to account for the increasing maternal age during the study period when examining the incidence of GBC. As our data comprised population data, we used all women giving birth during the study period as a standard population for the calculation of the indirect age-standardized rate.

Binary logistic regression models were used to identify independent factors associated with dichotomous outcomes. Analysis of neonatal outcomes was limited to singleton births due to the small number of multiple pregnancies (1.6%), the lack of data on the second baby in twin pregnancies and to avoid the confounding effect of multiple pregnancies [18]. These models incorporated all factors associated with each outcome in univariable analyses (p<0.20). Potential confounders including maternal age, parity, plurality, pre-existing chronic conditions such as diabetes and hypertension, previous CS, smoking during pregnancy, hospital sector (public or private) and remoteness of residence were also included. Odds ratio (OR), adjusted odds ratio (AOR), and 95% confidence interval (CI) were calculated and variables with a statistical level of significance (P-value) of <0.05 were considered significant. We consulted with clinicians in the research team, including an obstetrician, a breast surgeon, a neonatologist and an oncologist, to determine which interactions were plausible and limited our investigations to these. All variables in a regression models were assessed for collinearity with a variance inflation factor (VIF) threshold of <3. We tested the interaction term between pre-existing hypertension and smoking during pregnancy, and there was no evidence of interactions (Wald-test p>0.05). Data analysis was undertaken using Statistical Package for Social Sciences (SPSS) version 25.0 (IBM Corp, New York, United States).

## Ethics approval

NSW Population & Health Services Research Ethics Committee provided the ethical approval for the project (reference HREC/17/CIPHS/11).

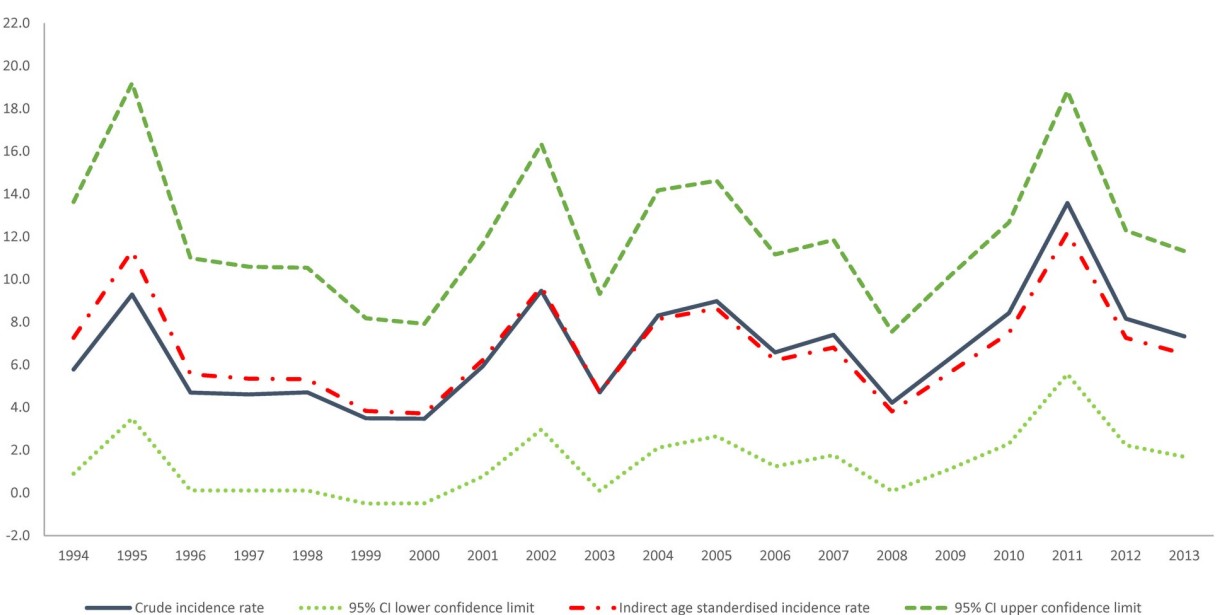

**Fig 2. Crude, indirect age-standardized incidence rate with upper and lower limits for 95%CI of the indirect age-standardized incidence rate of breast cancer diagnosis during pregnancy in NSW 1994–2013 per 100,000 women giving birth.**

## Results

There were 122 women with a first-time diagnosis of breast cancer during pregnancy (GBC group) and 1,782,994 women who gave birth without a history of any type of cancer before or during pregnancy (Fig 1).

### Incidence of GBC

The crude incidence rate of GBC in NSW from 1994 to 2013 was 6.8 diagnoses of GBC per 100,000 women giving birth (Fig 2). The incidence of GBC increased from 5.8 per 100,000 women giving birth in 1994 to 7.3 per 100,000 women giving birth in 2013, reaching a peak of 13.6 per 100,000 women giving birth in 2011. This represents an average annual increase of 2.8% (95%CI: -0.3%– 5.9%) per year. However, this increasing trend in incidence was not significant (p = 0.075).

### Maternal characteristics

**Age.** The mean (SD) age of women with GBC was significantly higher than that of women giving birth with no cancer (34.8 ± 4.4 years vs 29.6 ± 5.6 years, p<0.001; mean difference 5.27 years (95%CI 4.48–6.1)). The odds of GBC among women aged ≥35 years was six times the odds for those aged <35 years (AOR 6.09, 95%CI 4.02–9.20) (Table 1). Among women with GBC, there were 16 (13%) aged less than 30 years, 33 (27%) aged between 30 to 34 years, 54 (44%) between 35 to 39 and 19 (16%) aged 40 years or over.

**Timing of diagnosis and stage of cancer.** Of the 122 women with GBC, 25 (20.5%) were diagnosed in the first trimester, 39 (32.0%) in the second and 58 (47.0%) in the third trimester. Data on cancer stage were available for 113 (92.6%) and missing for 9 (7.4%). Of the 113 women with a known cancer stage, 42 (37.2%) were stage 1, 64 (56.6%) stages 2–3, and 7 (6.2%) stages 4 cancer.

**First-time mothers.** Women with GBC were less likely to be first-time mothers (nullipara) than women in the comparison group (32% vs 41.8%). However, when adjusting for age

**Table 1. Maternal characteristics and pre-existing conditions.**

| | Breast Cancer N = 122 | No cancer (N = 1,782,994) | OR (95% CI) | AOR (95% CI) |
|---|---|---|---|---|
| **Country of birth** | | | | |
| Other countries | 40(32.8) | 517504(29) | | Reference |
| Australia | 82(67.2) | 1260229(70.7) | 0.84 (0.58–1.23) | 1.17 (0.76–1.79) |
| Not stated* | 0(0) | 5261(0.3) | | |
| **Maternal age** | | | | |
| <35 | 49(40.2) | 1431525(80.3) | | Reference |
| = >35 | 73(59.8) | 350771(19.7) | 6.08 (4.23–8.73) | 6.16 (4.09–9.27) |
| Not stated* | 0(0) | 698(0) | | |
| **Parity** | | | | |
| Nullipara | 39(32) | 744494(41.8) | Reference | Reference |
| Para 1+ | 83(68) | 1036047(58.1) | 1.53 (1.05–2.24) | 1.08 (0.7–1.67) |
| Not stated* | 0(0) | 2453(0.1) | | |
| **Plurality** | | | | |
| Singleton | 120(98.4) | 1756474(98.5) | Reference | Reference |
| Multiple pregnancy | 2(1.6) | 26520(1.5) | 1.1 (0.27–4.47) | 0.52 (0.07–3.73) |
| **Previous CS** | | | | |
| No previous CS | 86(70.5) | 1369535(76.8) | Reference | Reference |
| CS 1+ | 18(14.8) | 193811(10.9) | 1.48 (0.89–2.46) | 0.97 (0.57–1.68) |
| Not stated* | 18(14.8) | 219648(12.3) | | |
| **Smoking during pregnancy** | | | | |
| No | 118(96.7) | 1502063(84.2) | Reference | Reference |
| Yes | 4(3.3) | 275928(15.5) | 0.18 (0.07–0.50) | 0.29 (0.11–0.79) |
| stated* | 0(0) | 5003(0.3) | | |
| **Pre-existing hypertension** | | | | |
| No | 118(96.7) | 1767310(99.1) | Reference | Reference |
| Yes | 4(3.3) | 15684(0.9) | 3.82 (1.41–10.35) | 2.43 (0.77–7.69) |
| **Pre-existing diabetes** | | | | |
| No | 122(100) | 1772966(99.4) | NA | NA |
| Yes | 0(0) | 10028(0.6) | NA | NA |
| **Remoteness** | | | | |
| Major Cities | 101(82.8) | 1354589(76) | Reference | Reference |
| Inner Regional | 17(13.9) | 308563(17.3) | 0.74 (0.44–1.24) | 0.99 (0.57–1.75) |
| Outer Regional | 3(2.5) | 88797(5) | 0.45 (0.14–1.43) | 0.74 (0.23–2.35) |
| Remote/very remote | 1(0.8) | 12583(0.7) | 1.07 (0.15–7.64) | 2.14 (0.3–15.47) |
| Not stated* | 0(0) | 18462(1) | | |

OR: crude odds ratio, AOR: adjusted odds ratio

*Not included in the analysis

#No previous birth.

and other maternal characteristics, the association was not significant (AOR 1.09, 95%CI: 0.70–1.68) (Table 1).

## Pregnancy complications and obstetric management (mode and timing of birth)

There were no significant differences in the rates of gestational diabetes, gestational hypertension or hospital transfer for women with or without GBC (Table 2).

**Table 2. Obstetric management and pregnancy complications by cancer status.**

| Outcome | Breast cancer | No cancer (reference) | OR(95% CI) | AOR(95% CI)* |
|---|---|---|---|---|
| **Induction of labour** | | | | |
| No | 29(23.8) | 1087440(61) | | |
| Yes | 51(41.8) | 438172(24.6) | 4.36 (2.77–6.89) | 4.40 (2.63–7.38) |
| Not applicable (Pre-labour CS)** | 42(34.4) | 256929(14.4) | | |
| Not stated* | 0(0) | 453(0) | | |
| **Induction of labour or pre-labour CS** | | | | |
| No | 29(23.8) | 1087440(61) | | |
| Yes | 93(76.2) | 695101(39) | 5.02 (3.31–7.61) | 4.96 (3.06–8.05) |
| *Preterm* | *48(51.6)* | *46855(6.7)* | | |
| *Term* | *45(48.4)* | *648116(93.2)* | | |
| *Not stated* | *0(0.0)* | *130(0.0)* | | |
| Not stated | 0(0) | 453(0) | | |
| **Mode of birth** | | | | |
| Vaginal birth*** | 67(54.9) | 1330464(74.6) | | |
| Birth By CS | 55(45.1) | 451638(25.3) | 2.42 (1.69–3.46) | 2.46 (1.57–3.86) |
| Not stated | 0(0) | 892(0.1) | | |
| **Gestational diabetes** | | | | |
| No | 117(95.9) | 1701488(95.4) | | |
| Yes | 5(4.1) | 81506(4.6) | 0.89 (0.36–2.18) | 0.57 (0.21–1.56) |
| **Gestational Hypertension** | | | | |
| No | 118(96.7) | 1674225(93.9) | | |
| Yes | 4(3.3) | 108769(6.1) | 0.52 (0.19–1.41) | 0.55 (0.20–1.51) |
| **Hospital sector** | | | | |
| Public | 86(70.5) | 1395153(78.2) | | |
| Private | 36(29.5) | 387799(21.7) | 1.51 (1.02–2.22) | 1.11 (0.72–1.73) |
| Not stated** | 0(0) | 42(0) | | |
| **Transferred to another hospital** | | | | |
| No | 118(96.7) | 1723181(96.6) | | |
| Yes | 4(3.3) | 59053(3.3) | 0.99 (0.37–2.68) | 1.40 (0.50–3.92) |
| Not stated** | 0(0) | 760(0) | | |

OR: crude odds ratio, AOR: adjusted odds ratio

*All variables are adjusted for maternal characteristics

**Not included in the analysis

***Including breech and instrumental birth.

**Birth intervention.** Among women with GBC, 51 (41.8%) had labour induction; of these, 41 (80.4%) had a vaginal birth and 10 (19.6%) gave birth by CS (Table 2). After adjusting for maternal characteristic, pre-existing conditions and hospital sector (public or private), women with GBC had significantly higher odds of labour induction (AOR 4.40, 95%CI 2.63–7.38) and CS (AOR 2.46, 95% CI 1.57–3.86) than women without cancer (Table 2).

**Labour induction and pre-labour CS.** Ninety-three (76.2%) women with GBC gave birth either by labour induction or pre-labour CS compared to 695,101 (39%) in the control group. After adjusting for maternal characteristics, pre-existing conditions and hospital sector, the odds of labour induction or pre-labour CS were significantly higher in the GBC group (AOR 4.96, 95% CI 3.06–7.79) (Table 2).

Among women who gave birth by labour induction or pre-labour CS, there was a higher rate of preterm birth in women with GBC (n = 48, 51.6%) compared to women with no cancer (n = 46,855, 6.7%) p<0.001.

**Timing of diagnosis, stage of cancer and birth by labour induction or pre-labour CS.** Among the 93 women with GBC who gave birth by labour induction or pre-labour CS, 10 (10.8%) were diagnosed in the first trimester, 32 (34.4%) in the second trimester, and 51 (54.8%) in the third trimester. Of those women diagnosed in the third trimester, 39 (76.5%) were diagnosed before 37 weeks gestation. Seven (70%) of the women diagnosed in the first trimester gave birth prematurely compared to 19 (59%) of the women diagnosed in the second trimester and 22 (56%) of the women diagnosed in the third trimester before 37 weeks gestation. However, the rate of preterm birth among women diagnosed in the second and third trimester (<37 weeks) was not significantly different from that in women diagnosed in the first trimester (OR 0.63, 95%CI: 0.14–2.88 and OR 0.55, 95%CI: 0.12–2.47 respectively) (S1 Table).

Among the 93 women with GBC who gave birth by labour induction or pre-labour CS, there were 27 (29.0%) with cancer stage 1, 54 (58.1%) with stages 2–3, and 6 (6.5%) with stages 4. Cancer stage was not known for 6 (6.5%). Fourteen (52%) of the women with stage 1 delivered prematurely compared to 31 (57%) of the women with stages 2–3 and 1 (17%) of the women with stages 4. The rate of preterm delivery among women with cancer stages 2–3 or stages 4 was not significantly different from that in women with cancer stage 1 (OR 1.25, 95% CI: 0.49–3.17 and OR 0.19 95%CI: 0.02–1.81 respectively) (S1 Table).

## Neonatal outcomes

The 122 pregnancies resulted in the birth of 120 singletons and two sets of twins. Table 3 describes the neonatal outcomes for 120 singleton babies born to women with GBC and 902,653 singleton babies born to women with no cancer. There were no stillbirths or neonatal deaths among babies born to women with GBC. Babies born to women with GBC were more likely to, require a high level of resuscitation including intermittent positive pressure respiration and external cardiac massage (11% vs 5%, AOR 2.01, 95%CI: 1.12–3.62). They are also more likely to be admitted to special care nursery (SCN) or neonatal intensive care unit (NICU) (42% vs 15% AOR, 3.74, 95%CI: 2.58–5.43) than babies born to women with no cancer. However, after adjusting for preterm birth, there is no significant difference in the odds of the need for a high level of resuscitation or admission to NICU/SCN between babies born to women with GBC versus those who were born to women with no cancer (AOR, 0.98 (95%CI: 0.54–1.81) and AOR,1.28 (95%CI: 0.81–2.02) respectively). Four neonates had major neonatal morbidities; one baby had congenital cardiomyopathy and three (34, 34 and 33 weeks' gestation) had respiratory distress syndrome of newborn, two of whom required prolonged ventilatory support.

The median gestational age at birth for babies born to women with GBC was lower than that for babies born to women with no cancer (37 weeks (IQR 35–38) vs 39 weeks (IQR 38–40), p < 0.001). The odds of preterm birth were higher in babies born to women with GBC (AOR 12.93, 95% CI 8.97–18.64).

The mean birthweight for live-born singletons to women with GBC was significantly lower than that for those born to women with no cancer (2,905 ± 634 g vs 3,409 ± 546 g, p<0.001). The birthweight distribution for preterm babies in both groups is shown in S2 Table.

**Preterm birth in babies born to women with GBC.** There were 53 (44.2%) preterm births among the 120 singletons born to women with GBC. Of these, 22 (42%) were born by induction of labour and 26 (49%) were born by pre-labour CS. Thirty-five (66%) of the preterm births were late preterm born between 34 and 36 weeks gestation, 17 (32.1%) were

**Table 3. Neonatal outcomes for singleton babies by maternal cancer status.**

| Outcome | Breast cancer | No cancer (reference) | OR(95% CI) | AOR(95% CI)* |
|---|---|---|---|---|
| **Sex of baby** | | | | |
| Male | 59(49.2) | 903557(51.4) | | |
| Female | 61(50.8) | 851876(48.5) | 1.10 (0.77–1.57) | 1.10 (0.77–1.57) |
| Indeterminate** | 0(0) | 222(0) | | |
| Not stated** | 0(0) | 819(0) | | |
| **Preterm birth** | | | | |
| No | 67(55.8) | 1654251(94.2) | | |
| Yes | 53(44.2) | 101834(5.8) | 12.85 (8.96–18.43) | 13.17 (9.14–18.96) |
| Not stated | 0(0) | 389(0) | | |
| **Small for gestation**\*** | | | | |
| No | 108(90) | 1568077(89.8) | | |
| Yes | 12(10) | 178297(10.2) | 0.98 (0.54–1.78) | 1.18 (0.65–2.16) |
| **Birthweight<2500 g**\*\* | | | | |
| No | 88(73.3) | 1668218(95.5) | | |
| Yes | 32(26.7) | 77482(4.4) | 7.85 (5.24–11.77) | 9.1 (6.02–13.77) |
| Not stated** | 0(0) | 674(0) | | |
| **5 min Apgar**\*\*\* | | | | |
| >7 | 115(95.8) | 1686705(96.6) | | |
| 7 or less | 5(5.8) | 54000(4.1) | 1.36 (0.56–3.33) | 1.32 (0.54–3.24) |
| Not stated | 0(0) | 5669(6.2) | | |
| **High resuscitation**\*\*\***#** | | | | |
| No | 86(71.7) | 1305741(74.8) | | |
| Yes | 13(10.8) | 91537(5.2) | 2.16 (1.20–3.87) | 2.01 (1.12–3.62) |
| Not stated** | 21(17.5) | 349096(20) | | |
| **Admitted to SCN/NICU for 4 hours or more**\*\*\* | | | | |
| No | 70(58.3) | 1480563(84.8) | | |
| Yes | 50(41.7) | 264646(15.2) | 4.00 (2.78–5.75) | 3.74 (2.58–5.43) |
| Not stated** | 0(0) | 1165(0.1) | | |
| **Discharge status** | | | | |
| Discharged | 113(94.2) | 1664307(94.8) | NA | NA |
| Stillborn | 0(0) | 10100(0.6) | | |
| Neonatal death | 0(0) | 3965(0.2) | | |
| Transferred | 7(5.8) | 77088(4.4) | | |
| Not stated | 0(0) | 1014(0.1) | | |

OR: crude odds ratio, AOR: adjusted odds ratio

*All variables are adjusted for maternal characteristics (5 min Apgar, High resuscitation and Admitted to SCN/NICU are also adjusted to the method of birth)

**Not included in the analysis

***live birth only

# intermittent positive pressure respiration and external cardiac massage

moderately preterm (32–33 weeks gestation), and 1 (1.9%) was early preterm (<32 weeks gestation) (S2 Table).

The mean birthweight of preterm babies born to women with GBC (2,469 ± 453 g) was significantly lower than term babies (3,250 ± 539 g) (p<0.001). Among the preterm babies of women with GBC, there were 29 (54.7%) babies with birthweight <2500 grams compared to 3 (4.5%) among term babies (p<0.001). However, among babies of women with GBC, the prevalence of SGA was lower for preterm compared with term babies (1.9% vs 18.8%, p = 0.004).

**Hospital admissions during the neonatal period.**   Of the 120 singletons born to women with GBC, hospitalization data were available for 102 (85%). Of these, 53 (52%) had at least one hospital admission within 28 days of birth (44 had one admission, 6 had two admissions, and 3 had three admissions).

## Discussion

We found an overall incidence of GBC in NSW between 1994 and 2013 of 6.8 per 100,000 women giving birth and that women with GBC had higher rates of planned preterm birth either by induction of labour or a pre-labour CS compared to women with no cancer. Surprisingly, the rate of planned preterm birth for women with GBC was not impacted by the timing of diagnosis or stage of cancer. Babies born to women with GBC were more likely to be preterm, require a high level of resuscitation and be admitted to SCN or NICU. In contrast, the proportion of SGA for preterm babies was very low at 1.9%, suggesting planned preterm birth for maternal management. There were no stillbirths or neonatal deaths among these babies, and the prevalence of major neonatal morbidities was relatively low.

Our results revealed a 20-year trend of increasing GBC incidence but that this trend was not statistically significant.

The odds of GBC were six times higher among women aged ≥35 years compared to those <35 years of age. Furthermore, women with GBC were significantly older than women with no cancer—women aged ≥35 years, comprised 59.8% of the GBC group compared to only 19.7% of the no cancer group.

In agreement with previous studies [3, 19, 20], our results show that women with GBC have higher rates of labour induction and/or delivery by CS than women with no cancer. It has been argued that the higher rates of labour induction and pre-labour CS in GBC are due to management decisions relating to the stage of cancer at diagnosis [21]. However, our results showed no differences in the odds of preterm labour induction or pre-labour CS among women diagnosed at different trimesters or with different stages of cancer at diagnosis. Nonetheless, these results should be interpreted with caution owing to the relatively low incidence rate of GBC and the small number of cases of GBC in this study. Additionally, as we were not able to obtain data relating to cancer treatment, we could not determine whether management decisions relating to treatment were associated with increased rates of labour induction and pre-labour CS.

There was a high rate of preterm birth among women with GBC. The majority of these births were planned and considered iatrogenic from a neonatal perspective. This is consistent with the high rate of preterm labour among women with GBC previously reported by Loibl and colleagues in 2012 where they argued that decisions to initiate early iatrogenic birth are often taken in the absence of a clear clinical indication [10]. In our study, we were unable to show any association between the high rate of iatrogenic preterm birth and any specific cancer stage or timing of diagnosis. This finding supports the views of Loibl and colleagues. However, owing to the small number of cases in our study, this finding should be treated with caution.

Preterm babies, whether born to women with GBC or to women with no cancer, have higher rates of adverse neonatal outcomes than babies born at ≥ 37 weeks [22]. Thus, in our cohort preterm babies of women with GBC had lower birthweights and increased rates of resuscitation and admission to SCN/NICU than term babies. Preparing, and then caring, for a preterm baby is demanding and inevitably places any mother at increased risk of anxiety and stress [23]. Women with GBC already experience high levels of fatigue and sleep disturbance underpinned by both the side effects of chemotherapy (whether given during pregnancy or after birth) and psychological and biological factors [24–26]. Caring for a preterm baby who

has increased needs is likely to present unique challenges to mothers coping with cancer symptoms in parallel with treatment side effects. These factors, together with the potential longer-term impacts of low birthweight and neonatal complications associated with premature birth, suggest that decisions regarding pre-term induction of labour or pre-labour CS in women with GBC needs to be carefully considered. A delicate balance is required to be drawn between any potential benefits to the mother of commencing early cancer treatment and the potential adverse effects of preterm birth both to the mother and her child.

Current literature suggests that having gestational cancer or cancer treatment during pregnancy may be associated with a higher risk of SGA [6, 10]. However, our results do not support this, possibly due to the small sample size of women with GBC which may have impacted the power of the study to detect any such difference.

Although our data show a low prevalence of major neonatal morbidities in our preterm babies, we did not have data to examine the long-term developmental effect of these infants. Amant and colleagues (2015) previously found that preterm babies born to women with a diagnosis of cancer during pregnancy are more likely to have long term developmental problems whether or not they were exposed to chemotherapy while in utero [7]. Given this, it is important to promote a term birth where clinically possible in order to avoid the potential negative effects of preterm birth on both the mother and her baby.

### Strengths and limitations

An important strength of our study was the population-based design that included all births in NSW over a 20-year period from which we identified all women with invasive breast cancer during pregnancy. Limitations include the lack of information on Indigenous status, and breast cancer treatment. The latter has limited our ability to examine the maternal and perinatal outcomes by exposure to treatment. The third limitation was resultant from the fact that, early pregnancy loss before 20 weeks' gestation is out of the scope of the perinatal data collection.

### Conclusions

The odds of GBC were six times higher among women aged ≥35 years compared to those <35 years of age. There was a high rate of preterm birth among women with GBC, which could not be explained by the timing of breast cancer diagnosis or stage of cancer at diagnosis. Spontaneous onset of labour of preterm birth was uncommon with most births following induction of labour or pre-labour CS. This had a minimal impact in the short-term as major neonatal morbidity was uncommon.

### Supporting information

**S1 Table. Timing of diagnosis and stage of cancer by gestational age at birth for the 93 women who gave birth by induction of labour or pre-labour CS.**
(DOCX)

**S2 Table. Characteristics of singleton preterm babies by cancer status.**
(DOCX)

**S3 Table. Regression model of Table 1.**
(DOCX)

**S4 Table. Regression models of Table 2.**
(DOCX)

**S5 Table. Regression models of Table 3.**
(DOCX)

## Author Contributions

**Conceptualization:** Nadom Safi, Christobel Saunders, Andrew Hayen, Antoinette Anazodo, Kei Lui, Zhuoyang Li, Marc Remond, Michael Nicholl, Alex Y. Wang, Elizabeth Sullivan.

**Formal analysis:** Nadom Safi, Zhuoyang Li.

**Funding acquisition:** Elizabeth Sullivan.

**Methodology:** Nadom Safi, Christobel Saunders, Andrew Hayen, Antoinette Anazodo, Kei Lui, Zhuoyang Li, Marc Remond, Michael Nicholl, Alex Y. Wang, Elizabeth Sullivan.

**Supervision:** Andrew Hayen, Antoinette Anazodo, Alex Y. Wang, Elizabeth Sullivan.

**Validation:** Nadom Safi.

**Writing – original draft:** Nadom Safi.

**Writing – review & editing:** Nadom Safi, Christobel Saunders, Andrew Hayen, Antoinette Anazodo, Kei Lui, Zhuoyang Li, Marc Remond, Michael Nicholl, Alex Y. Wang, Elizabeth Sullivan.

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
