## [Decision Letter · Decision Letter 0]

4 Nov 2020

PONE-D-20-16908

Gestational Breast Cancer in New South Wales: A population-based linkage study of incidence, management, and outcomes.

PLOS ONE

Dear Dr. Sullivan,

Thank you for submitting your manuscript to PLOS ONE. After careful consideration, we feel that it has merit but does not fully meet PLOS ONE’s publication criteria as it currently stands. Therefore, we invite you to submit a revised version of the manuscript that addresses the points raised during the review process.

We look forward to receiving your revised manuscript.

Kind regards,

Angela Lupattelli, PhD

Academic Editor

PLOS ONE

Reviewers' comments:

Reviewer's Responses to Questions

**Comments to the Author**

1. Is the manuscript technically sound, and do the data support the conclusions?

Reviewer #1: Partly

Reviewer #2: Yes

2. Has the statistical analysis been performed appropriately and rigorously? 

Reviewer #1: Yes

Reviewer #2: Yes

3. Have the authors made all data underlying the findings in their manuscript fully available?

Reviewer #1: Yes

Reviewer #2: Yes

4. Is the manuscript presented in an intelligible fashion and written in standard English?

Reviewer #1: Yes

Reviewer #2: Yes

5. Review Comments to the Author

Reviewer #1: This population-based study describes the obstetrical outcome of women with a breast cancer diagnosis during pregnancy in . Although there are already many series published on this topic, the novelty of this manuscript is the inclusion of oncological data (cancer stage). An important limitation is the lack of treatment information, as treatment might be a major confounder of outcomes for mother and child.

I have some major revisions and minor suggestions:

Methods:

Why only include pregnancies that ended in live birth or stillbirth? Not including pregnancies ending in an early miscarriage or termination will underestimate the incidence. This should be mentioned in discussion.

Staging: was this according to a standardized classification? Were records reviewed or was this already registered as such in the database? How reliable are these data (was proper staging during pregnancy performed)?

Page 7: Odds ratio (OR), adjusted odds ratio (AOR), and 95% confidence interval (CI) were calculated and variables with a statistical level of significance (P-value) of <0.05….. > misses a part?

Consulted with clinicians in the research teams > what kind of clinicians? Obstetricians? Oncologists?

Results:

Birth interventions / Labour induction and pre-labour CS > why first mentioning only induction and afterwards both induction and pre-labour CS together?

Neonatal outcomes (p12): Babies born to women with GBC were more likely to, require

a high level of resuscitation including intermittent positive pressure respiration and external cardiac

massage (11% vs 5%, AOR 2.01, 95%CI: 1.12 – 3.62). Probably this was related to prematurity? What are the differences when corrected for prematurity? Also, hospital admissions during the neonatal period (p14): When corrected for preterm birth, is this still significantly different compared to the no cancer group?

Page 13: The median gestational age at birth for babies born to women with GBC was lower than babes born > babes = babies

Preterm birth in babies born to women with GBC (p13): it is normal that birthweight of preterm infants are lower compared to term infants; it is more interesting to look at the birthweight percentiles. (birthweight adjusted for gestational age at delivery, and other factors) � Discussion p15; Thus, preterm babies of women with GBC had lower birthweights and increased rates of resuscitation and admission to SCN/NICU than term babies of women with GBC. I think this is so for all preterm babies, not only for the babies born from women with breast cancer; please rephrase.

Discussion:

It has been argued that the higher rates of labour induction and pre-labour CS in GBC are due to management decisions relating to the stage of cancer at diagnosis (19). > But there is no information available about treatment during pregnancy / postpartum for this GBC group, so could the higher amount of labour induction / pre-term CS still be because of management decisions regardless of stage of cancer?

Suggestions:

Page 4, alinea 1: Suggestion to mention the first most common cancer during pregnancy in Australia?

Page 4 Alinea 2: Suggestion to mention preterm birth as a risk factor for developmental problems, irrespective of whether or not the baby is born to a women with GBC?

Page 7: All variables in a regression models were assessed > or “in the regression models” or “in a regression model”

Page 11: p<0.001 in ( )

Page 12: Maybe mention the 2 multiple pregnancies before continuing with the singletons/or separately

Page 12: Two of those with respiratory distress syndrome were born at 34 weeks gestation and one at 33 weeks gestation. Already mentioned before in ( ), leave that part out, or this sentence.

Discussion: Last alinea (p14-15): I suggest to further comment on this. Probably the cancer diagnosis itself, the (clinical, psychological) stress and/or the higher incidence of preterm deliveries might explain the higher incidence of planned deliveries?

Strengths and limitations: I suggest to go further into detail about the lack of information in respect to the treatment.

Table 2: is it possible to add preterm birth (for all patients, and for patients with IOL specifically?, as mentioned in the results section)

Table 3: Interestingly, SGA is not different for patients with breast cancer vs no cancer; please comment on this in discussion as in current literature cancer/cancer treatment seems to be associated with a higher risk of SGA. In this dataset, there is no information on treatment during pregnancy, also the ‘non cancer’ patients include potentially high risk patients for SGA (systemic disease?). What about ethnicity in this population (were all patients Australian, Caucasian?)? Still, the incidence (10%) is comparable to the expected rates in the general population.

Reviewer #2: Criteria:

This study meets all of PLOSONE’s basic review criteria.

Additional comments:

This is a very crucial study and I’m glad that there is a possible addition to the body of knowledge here. Particularly interesting to see that there were no stillbirths or infant deaths, though other studies like Loibl et al’s only had 2 out of ~450 so perhaps it is keeping with the overall trend. The sample size is relatively small, and you do mention that this might affect outcomes in one section, but it might be good to put a more prominent clarification in the methods section or in limitations.

There are some language use issues that I would recommend be further considered. Specifically, even though current cancer registries do not collect good data regarding gender (or any, in some cases), it is pertinent to be as accurate as possible in describing your participants. I would recommend directly stating that we are most likely talking about cisgender women here, even if the actual data does not exist. This is particularly pertinent since there is a growing call for this kind of specificity and consideration within oncology (see: Lucille Kerr et al. “TRANScending discrimination in health & cancer care”). This paper would also benefit from consistency around “developed”/”high income” countries. “High income” is preferred in most settings, and would be particularly useful here since part of the consideration throughout should be about access to resources (see: comment on remoteness below).

The final point that may need some refinement here is the demographic data. It would have been good to at least see Aboriginal heritage being flagged – the NSWCR does recommend dealing with Aboriginal status with care in analysis, but there is still a need to mention it, particularly when there is discussion of diabetes which does significantly affect those populations. This is doubly so when you take into account the number of Aboriginal people living remotely, which also seemed to be another significant factor which did not get discussed here. Distance from care facilities and professionals plays a significant role in maternal and natal health, and though your sample size may have been too small to properly capture any effects, it is at least worth noting. Absence is its own kinda data! I think that your paper may benefit from providing at least a little bit of this kind of demographic/sociological lens, in part so that it can become a relevant resource for a wider audience. But an otherwise good paper!

6. PLOS authors have the option to publish the peer review history of their article (what does this mean?). If published, this will include your full peer review and any attached files.

Reviewer #1: No

Reviewer #2: No

---

## [Author Response · Author response to Decision Letter 0]

20 Dec 2020

Joerg Heber 

Editor-in-Chief 

PLOS ONE 

11 December 2020 

Dear Professor Heber,

RE: PONE-D-20-16908 Gestational Breast Cancer in New South Wales: A population-based linkage study of incidence, management, and outcomes.

Thank you for the opportunity to submit the revised version of the manuscript PONE-D-20-16908 entitled “Gestational Breast Cancer in New South Wales: A population-based linkage study of incidence, management, and outcomes” for consideration of publication in PLOS ONE.

We appreciate having the opportunity to respond to the comments from the reviewers. We have addressed these comments and attached clean and marked up versions of the manuscript to reflect text changes. The responses to specific comments are provided below.

Reviewer #1:

Methods:

Why only include pregnancies that ended in live birth or stillbirth? Not including pregnancies ending in an early miscarriage or termination will underestimate the incidence. This should be mentioned in discussion.

Authors:

The NSW perinatal data collection does not include information on early pregnancy loss and for this reason we were not able to include information about pregnancies ending in early miscarriage or termination. We have updated the limitations section to read: “Limitations include the lack of information on Indigenous status and breast cancer treatment. The latter has limited our ability to examine the maternal and perinatal outcomes by exposure to treatment. The third limitation was resultant from the fact that early pregnancy loss before 20 weeks’ gestation is out of the scope of the perinatal data collection.”

Reviewer #1:

Staging: was this according to a standardized classification? Were records reviewed or was this already registered as such in the database? How reliable are these data (was proper staging during pregnancy performed)?

Authors:

The staging is based on data provided by NSWCR which was originally available as: stage 0 - carcinoma in situ; stage 1 - cancer localized to the tissue of origin; stage 2 - cancer that has spread to the regional lymph nodes and/or adjacent organs; and stage 3 - distant metastasis. We modified these staging categories to reconcile them with the Royal College of Pathologists of Australasia cancer staging classification (i.e. recategorizing NSWCR stage 2 as RACP stages 2-3, and NSWCR stage 3 as RACP stage 4). This is now further described in the methods section.

Reviewer #1:

Page 7: Odds ratio (OR), adjusted odds ratio (AOR), and 95% confidence interval (CI) were calculated and variables with a statistical level of significance (P-value) of <0.05….. > misses a part?

Authors:

We have updated the sentence to read: Odds ratios (OR), adjusted odds ratios (AOR), and 95% confidence intervals (CI) were calculated and variables with a statistical level of significance (P-value) of <0.05 were considered significant.

Reviewer #1:

Consulted with clinicians in the research teams > what kind of clinicians? Obstetricians? Oncologists?

Authors:

We have added the specialities and updated the sentence to read; We consulted with clinicians in the research team including an obstetrician, a breast surgeon, a neonatologist and an oncologist.

Reviewer #1:

Birth interventions / Labour induction and pre-labour CS > why first mentioning only induction and afterwards both induction and pre-labour CS together?.

Authors:

Data relating to labour induction and pre-labour CS were first presented separately with comparison statistics being calculated to compare these outcomes between women with breast cancer and those with no cancer. These variables were then combined into a composite variable as a means of gaining a clearer understanding of the likelihood of planned deliveries (whether by pre-labour CS or induction), again with a comparison between women with GBC and those with no cancer.

Reviewer #1:

Neonatal outcomes (p12): Babies born to women with GBC were more likely to, require a high level of resuscitation including intermittent positive pressure respiration and external cardiac massage (11% vs 5%, AOR 2.01, 95%CI: 1.12 – 3.62). Probably this was related to prematurity? What are the differences when corrected for prematurity? Also, hospital admissions during the neonatal period (p14): When corrected for preterm birth, is this still significantly different compared to the no cancer group?

Authors:

We have performed the analysis suggested by the reviewer to adjust for prematurity. We have added the following sentence to the results section: However, after adjusting for preterm birth, there is no significant difference in the odds of the need for a high level of resuscitation or admission to NICU/SCN between babies born to women with GBC versus those who were born to women with no cancer (AOR, 0.98 (95%CI:0.54-1.81) and AOR,1.28 (95%CI:0.81-2.02) respectively).

Reviewer #1:

Page 13: The median gestational age at birth for babies born to women with GBC was lower than babes born > babes =babies 

Authors:

We have corrected this sentence to read: The median gestational age at birth for babies born to women with GBC was lower than that of babies born to women with no cancer.

Reviewer #1:

Preterm birth in babies born to women with GBC (p13): it is normal that birthweight of preterm infants are lower compared to term infants; it is more interesting to look at the birthweight percentiles. (birthweight adjusted for gestational age at delivery, and other factors).

Authors:

The manuscript already addresses this issue by reporting information about babies who were small-for-gestational-age (SGA). These SGA results present the proportion of babies that were born who weighed below the 10% birthweight percentile for their gestational age (i.e. this SGA determination implicitly adjusts for prematurity/gestational age at delivery and other factors).

Reviewer #1:

Discussion p15; Thus, preterm babies of women with GBC had lower birthweights and increased rates of resuscitation and admission to SCN/NICU than term babies of women with GBC. I think this is so for all preterm babies, not only for the babies born from women with breast cancer; please rephrase.

Authors:

This section has been amended to address the reviewer’s comment. Thus, the rephrased text makes it clearer that preterm birth places increased anxiety and stress on any mother, whether or not they had GBC. However, it also clarifies that mothers with GBC experience additional issues beyond those of mothers with no cancer such as a fear of not seeing their baby grow up due to cancer-related death and the side effects of chemotherapy (whether given during pregnancy or after birth) and psychological and biological factors. The cumulative effect on mothers with GBC who have a preterm birth mean that careful consideration is required in respect of any decisions regarding preterm induction of labour or preterm CS. 

Reviewer #1:

It has been argued that the higher rates of labour induction and pre-labour CS in GBC are due to management decisions relating to the stage of cancer at diagnosis (19). > But there is no information available about treatment during pregnancy / postpartum for this GBC group, so could the higher amount of labour induction / pre-term CS still be because of management decisions regardless of stage of cancer?

Authors:

We agree with the reviewer’s comment, but data relating to breast cancer treatment were not available in the population dataset used in this study. Therefore our lead in sentence only refers to: “management decisions relating to the stage of cancer at diagnosis”. We do not refer to management decisions relating to treatment. To make the paragraph clearer, we have included the following statement at the end: “Additionally, as we were not able to obtain data relating to cancer treatment, we could not determine whether management decisions relating to treatment were associated with increased rates of labour induction and pre-labour CS.”

Reviewer #1:

Page 4, line 1: Suggestion to mention the first most common cancer during pregnancy in Australia?

Authors:

We have added information about melanoma, the most common cancer during pregnancy in Australia. The sentence now reads “In Australia, breast cancer is the second most common cancer diagnosed during pregnancy, after melanoma, with an incidence of 7.3 per 100,000 women giving birth.”

Reviewer #1:

Page 4 line 2: Suggestion to mention preterm birth as a risk factor for developmental problems, irrespective of whether or not the baby is born to a women with GBC?

Authors:

We added the following: It has been reported that preterm birth is a risk factor for developmental problems, irrespective of whether or not a baby is born to a women with GBC (8, 9).

8. Yeo KT, Safi N, Wang YA, Marsney RL, Schindler T, Bolisetty S, et al. Prediction of outcomes of extremely low gestational age newborns in Australia and New Zealand. BMJ Paediatr Open. 2017;1(1):e000205.

9. Paules C, Pueyo V, Martí E, de Vilchez S, Burd I, Calvo P, et al. Threatened preterm labor is a risk factor for impaired cognitive development in early childhood. American journal of obstetrics and gynecology. 2017;216(2):157.e1-.e7.

Reviewer #1:

Page 7: All variables in a regression models were assessed > or “in the regression models” or “in a regression model”

Authors:

Corrected to now read, “All variables in the regression models”

Reviewer #1:

Page 11: p<0.001 in ( )

Authors:

In text only results. 

Reviewer #1:

Page 12: Maybe mention the 2 multiple pregnancies before continuing with the singletons/or separately.

 Authors:

We have added the following: The 122 pregnancies resulted in the birth of 120 singletons and two sets of twins.

Reviewer #1:

Page 12: Two of those with respiratory distress syndrome were born at 34 weeks gestation and one at 33 weeks gestation. Already mentioned before in ( ), leave that part out, or this sentence.

Authors:

Deleted

Reviewer #1:

Discussion: Last line (p14-15): I suggest to further comment on this. Probably the cancer diagnosis itself, the (clinical, psychological) stress and/or the higher incidence of preterm deliveries might explain the higher incidence of planned deliveries?

Authors:

Whilst it is possible that maternal psychological distress resulting from a diagnosis of GBC may impact on decisions to undertake a planned delivery, our study does not have information available on this. We have data from a binational prospective observational study of women with GBC suggesting that the timing and mode of delivery for women with GBC was only due to maternal request/choice in 2.5% of cases. This data forms part of a manuscript of this separate study that is currently under review and so, unfortunately, cannot be cited in the current paper.

Reviewer #1:

Strengths and limitations: I suggest to go further into detail about the lack of information in respect to the treatment.

Authors:

This has been done.

Reviewer #1:

Table 2: is it possible to add preterm birth (for all patients, and for patients with IOL specifically?, as mentioned in the results section).

Authors:

In Table 2 we have added a subcategory of preterm and term delivery for those with “yes” under the composite variable “Induction of labour or pre-labour CS.”

Reviewer #1:

Table 3: Interestingly, SGA is not different for patients with breast cancer vs no cancer; please comment on this in discussion as in current literature cancer/cancer treatment seems to be associated with a higher risk of SGA.

Authors:

We are unable to comment on why SGA is not different for patients with breast cancer vs no cancer due to the small sample size. We have now included the sentence: “Current literature suggests that having gestational cancer or cancer treatment during pregnancy may be associated with a higher risk of SGA (6, 10). However, our results do not support this, possibly due to the small sample size of women with GBC which may have impacted the power of the study to detect any such difference”.

6. Van Calsteren K, Heyns L, De Smet F, Van Eycken L, Gziri MM, Van Gemert W, et al. Cancer during pregnancy: an analysis of 215 patients emphasizing the obstetrical and the neonatal outcomes. J Clin Oncol. 2010;28(4):683-9.

10. Loibl S, Han SN, von Minckwitz G, Bontenbal M, Ring A, Giermek J, et al. Treatment of breast cancer during pregnancy: an observational study. The Lancet Oncology. 2012;13(9):887-96.

Reviewer #1:

In this dataset, there is no information on treatment during pregnancy, also the ‘non cancer’ patients include potentially high risk patients for SGA (systemic disease?). 

Authors:

In relation to patients with systemic diseases, we were unable to demonstrate any difference in the rate of diabetes or hypertension between women with GBC and those with “no cancer”.

Reviewer #1:

What about ethnicity in this population (were all patients Australian, Caucasian?)?

Still, the incidence (10%) is comparable to the expected rates in the general population.

Authors:

Australian data do not provide information on ethnicity (Caucasian, Asian etc). However, they do provide information on the country of birth which is presented in Table 1.

Reviewer #2:

There are some language use issues that I would recommend be further considered. Specifically, even though current cancer registries do not collect good data regarding gender (or any, in some cases), it is pertinent to be as accurate as possible in describing your participants. I would recommend directly stating that we are most likely talking about cisgender women here, even if the actual data does not exist. This is particularly pertinent since there is a growing call for this kind of specificity and consideration within oncology (see: Lucille Kerr et al. “TRANScending discrimination in health & cancer care”). 

Authors:

In reference to gender-identity, we note that this is an important current issue. However, as this study only includes data relating to women who give birth, it implicitly is restricted to women who are cisgender. 

Reviewer #2:

This paper would also benefit from consistency around “developed”/”high income” countries. “High income” is preferred in most settings, and would be particularly useful here since part of the consideration throughout should be about access to resources (see: comment on remoteness below).

Authors:

“Developed” countries has been changed into “high-income” countries.

Reviewer #2:

The final point that may need some refinement here is the demographic data. It would have been good to at least see Aboriginal heritage being flagged – the NSWCR does recommend dealing with Aboriginal status with care in analysis, but there is still a need to mention it, particularly when there is discussion of diabetes which does significantly affect those populations. This is doubly so when you take into account the number of Aboriginal people living remotely, which also seemed to be another significant factor which did not get discussed here. Distance from care facilities and professionals plays a significant role in maternal and natal health, and though your sample size may have been too small to properly capture any effects, it is at least worth noting. Absence is its own kinda data!

Authors:

The information on Indigenous status is not available in our research dataset. We have updated the limitations section to read: “Limitations include the lack of information on Indigenous status and breast cancer treatment. The latter has limited our ability to examine the maternal and perinatal outcomes by exposure to treatment. The third limitation was resultant from the fact that early pregnancy loss before 20 weeks’ gestation is out of the scope of the perinatal data collection”.

Yours sincerely,

Professor Elizabeth Sullivan MD, MPH, MMed, MBBS, FAFPHM

Faculty of Health and Medicine Faculty of Health and Medicine, The University of Newcastle.

T (02) 4985 4355

E E.Sullivan@newcastle.edu.au

---

## [Editor Report · Decision Letter 1]

2 Jan 2021

Gestational Breast Cancer in New South Wales: A population-based linkage study of incidence, management, and outcomes.

PONE-D-20-16908R1

Dear Dr. Sullivan,

We’re pleased to inform you that your manuscript has been judged scientifically suitable for publication and will be formally accepted for publication once it meets all outstanding technical requirements.

Kind regards,

Angela Lupattelli, PhD

Academic Editor

PLOS ONE

---

## [Editor Report · Acceptance letter]

13 Jan 2021

PONE-D-20-16908R1 

Gestational Breast Cancer in New South Wales: A population-based linkage study of incidence, management, and outcomes. 

Dear Dr. Sullivan:

I'm pleased to inform you that your manuscript has been deemed suitable for publication in PLOS ONE. Congratulations! Your manuscript is now with our production department. 

Kind regards, 

on behalf of

Dr. Angela Lupattelli 

Academic Editor

PLOS ONE